# An online global survey and follow-up expert groups on the scope and needs related to training, research, and mentorship among early-career addiction medicine professionals

## Research Article

addiction psychiatry; early career addiction medicine professionals; mentorship; training assessment; substance use disorders

**Corresponding author:**
Roshan Bhad;
Email: roshan@aiims.edu

†ISAM NExT Consortium included 26 members across 22 countries.

Roshan Bhad[1,2] ⬤, Sophia Achab[2,3] ⬤, Parnian Rafei[2,4], Preethy Kathiresan[5], Hossein Mohaddes Ardabili[2,6], Jenna Butner[2,7], Laura Orsolini[2,8], Katrine Melby[2,9], Mehdi Farokhnia[2,10,11], Venkata Lakshmi Narasimha[2,12], Kelly Ridley[2,13], Serenella Tolomeo[2,14,15], Mitika Kanabar[2,16], Beatrice Matanje[2,17], Paolo Grandinetti[18], ISAM NExT Consortium[†], Marc Potenza[7], Hamed Ekhtiari[19,20] and Alexander Baldacchino[21]

[1]Department of Psychiatry & National Drug Dependence Treatment Centre (NDDTC), All India Institute of Medical Sciences (AIIMS), New Delhi, India; [2]Member of ISAM NExT (New Professionals Exploration, Training & Education Committee), International Society of Addiction Medicine (ISAM), Calgary, AB, Canada; [3]Psychological and Sociological Research Unit, Faculty of Medicine, University of Geneva, Geneva, Switzerland; [4]Department of Psychology, Faculty of Psychology and Education, University of Tehran, Tehran, Iran; [5]Department of Psychiatry, All India Institute of Medical Sciences (AIIMS), Jodhpur, India; [6]Psychiatry and Behavioural Sciences Research Center, Mashhad University of Medical Sciences, Mashhad, Iran; [7]Yale School of Medicine, New Haven, CT, USA; [8]Unit of Clinical Psychiatry, Department of Neurosciences/DIMSC, School of Medicine and Surgery, Polytechnic University of Marche, Ancona, Italy; [9]Department of Clinical Pharmacology, St. Olav University Hospital, Trondheim, Norway; [10]National Institute on Drug Abuse, National Institutes of Health (NIH), Baltimore, MD, USA; [11]National Institute on Alcohol Abuse and Alcoholism, National Institutes of Health (NIH), Baltimore, MD, USA; [12]Department of Psychiatry, All India Institute of Medical Sciences (AIIMS), Deoghar, India; [13]The Rural Clinical School of Western Australia, University of Western Australia, Albany, WA, Australia; [14]Institute of High Performance Computing, Agency for Science, Technology and Research (A-STAR), Singapore, Singapore; [15]Department of Pharmacology, Yong Loo Lin School of Medicine, National University of Singapore, Singapore; [16]Southern California Permanente Medical Group, Pasadena, CA, USA; [17]Lighthouse Trust, Lilongwe, Malawi; [18]Network of Early Career Professionals working in the area of Addiction Medicine (NECPAM) Seligenstadt, Germany; [19]Laureate Institute for Brain Research, Tulsa, OK, USA; [20]Department of Psychiatry, University of Minnesota, Minneapolis, MN, USA and [21]Division of Population and Behavioural Science, Medical School, University of St Andrews, St Andrews, UK

## Abstract

Addiction medicine is a rapidly growing field with many young professionals seeking careers in this field. However, early-career professionals (ECPs) face challenges such as a lack of competency-based training due to a shortage of trainers, limited resources, limited mentorship opportunities, and establishment of suitable research areas. The International Society of Addiction Medicine (ISAM) New Professionals Exploration, Training & Education (NExT) committee, a global platform for early-career addiction medicine professionals (ECAMPs), conducted a two-phase online survey using a modified Delphi-based approach among ECAMPs across 56 countries to assess the need for standardized training, research opportunities, and mentorship. A total of 110 respondents participated in Phase I (online key informant survey), and 28 respondents participated in Phase II (online expert group discussions on three themes identified in Phase I). The survey found that there is a lack of standardized training, structured mentorship programs, research funding, and research opportunities in addiction medicine for ECAMPs. There is a need for standardized training programs, improving research opportunities, and effective mentorship programs to promote the next generation of addiction medicine professionals and further development in the entire field. The efforts of ISAM NExT are well-received and give a template of how this gap can be addressed.

## Impact statement

This article is intended to assess and understand addiction medicine training, mentorship needs, and research opportunities among early-career addiction medicine professionals (ECAMPs) globally. Substance use disorders and addictive behaviors are growing public health problems across the globe, there is an increase in the burden of disease due to substance use disorders and addictive behaviors, yet there is a significant treatment gap across countries. Bridging the treatment gap required multiple strategies including workforce development. There are hardly any studies focusing on ECAMPs understanding their training, mentorship,

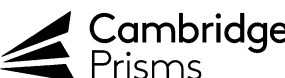



and research needs. This is the first of its kind major global study across 50 countries using both online surveys as well as expert group discussions to answer these questions. The target group studied was ECAMPs. We believe the group having recently undergone training or currently undergoing training in addiction medicine, their perspective is likely to give a realistic snapshot of the current status of addiction medicine training in the world. We have discussed the findings on the lack of standardized training, training needs, absence of robust, effective mentorship programs in most countries, and lack of research opportunities for this group except few developed countries. Despite some limitations, the present study gives insight into the current status of addiction medicine/psychiatry training across the globe for the point of ECAMPs. We believe that these challenges can be addressed using a collaborative approach with the support of global agencies working for workforce development in the health sector. The next generation of professionals can be better prepared and trained for the emerging global public health problem of substance use disorders and addictive behaviors. The intended impact is therefore for workforce development for future addiction medicine services.

## Introduction

Addiction medicine is a relatively developing field of medicine, with a growing number of early-career professionals (ECPs) opting for a career in this area (De Jong and Van De Wetering, 2009; Smith, 2011). However, there are several challenges in terms of the lack of well-structured training shortage of institutes with infrastructure for adequate training and trainers, that is, formally trained mental health and medical professionals (including faculty, mentors, or preceptors) with experience in addiction medicine (Soyka and Gorelick, 2009; Smith, 2011). These limitations prevent ECPs in several countries from pursuing a career in addiction medicine. There are limited resources and training opportunities for ECPs in upper-middle (UMICs) and lower-middle-income countries (LMICs). In high-income countries (HICs), where there is no or less dearth of experts and infrastructure, the challenges include receiving appropriate mentorship and choosing a suitable research area (De Jong and Van De Wetering, 2009; Ayu et al., 2015; Klimas et al., 2017). Therefore, there is a need for a global platform helping early-career addiction medicine professionals (ECAMPs), including trainees, and connecting them with each other and with trainers and mentors worldwide. There is also a need to facilitate the launch and implementation of standardized training programs, creating research and education opportunities, as well as fellowships and mentorship programs in each subspecialty of the addiction medicine field (Ayu et al., 2017). These needs become extremely necessary mainly due to the significant variability in the standards and quality of training programs in the field of addiction and/or psychiatry globally, which is a major challenge for many ECAMPs in many countries (Haber, 2011).

Similarly, the assessment of training in addiction medicine and/or psychiatry is limited. For example, the International Certification in Addiction Medicine by the International Society of Addiction Medicine (ISAM) is one of the few examples of a well-established association able to provide global standards in validating and certifying knowledge in addiction medicine for professionals. However, currently, the examination by ISAM includes assessment for theory based on multiple-choice questions (MCQs), without any practical exam or real case vignettes (el-Guebaly and Violato, 2011; Rasyidi et al., 2012). Hence, it is essential first to identify and clearly understand the needs and the demand for a standardized assessment of training in addiction medicine and/or psychiatry to develop and implement a universal curriculum in addiction training programs.

The ISAM NExT (New Professionals Exploration, Training & Education) committee was established in 2020 with the primary objective of increasing and improving the capacity of addiction medicine training and other educational activities among ECAMPs. The committee constitutes 30 early-career addiction professional members, including a chair, two co-chairs and members from 22 countries.

Building research collaboration across the globe and developing a practice-based research network is of high importance, given the eclectic nature of the field of addiction medicine and its significance is emphasized by policymakers and various global organizations, as well. Unfortunately, there are hardly any international organizations and networks in addiction medicine that address the need for researchers on a global platform for research collaboration. The National Institute on Drug Abuse (NIDA) is an organization that provides research support for early and mid-career addiction professionals; however, support is often limited to United States-based researchers and institutes (National Institute on Drug Abuse (NIDA), 2020). Assessing the need and scope for research opportunities exclusively for ECAMPs will inform policymakers regarding various issues and challenges. Quality mentoring and strategic planning, along with a favorable environment, are some of the elements that should be combined to create a successful career in research (Zachary, 2011; Alford et al., 2018) Moreover, there is a need to assess deficiencies in training, research interest, and need for mentoring among early-career addiction professionals and address important issues that may help them in career development to mid-career. This may motivate and encourage ECAMPs to take up addiction medicine as an informed career choice since they can see the career trajectory and growth prospects ahead. We conducted a two-phase global cross-sectional online survey among ECAMPs to understand the need and scope for standardized training, research opportunities, and mentorship in the field of addiction medicine.

## Methods

### Study design

A two-phase global online survey was conducted using a mixed-method, modified Delphi-based approach (McMillan et al., 2016; Niederberger and Spranger, 2020). The first phase of the survey was carried out using an online survey in which 270 ECAMPs were approached. The reason for conducting survey among ECAMPs was due to their unique position of undergoing training or recently completed the training in particular country giving insight into training need for addiction medicine and related issues of research, mentorship from respective countries during early phase of career. The first round of the survey took place from October 2020 to March 2021, and the results were finalized in April 2021. The second phase of the study comprised three focus group discussions to obtain consensus on key themes elicited in the first phase.

An online Google survey tool was prepared by the research team for phase I (available as Supplementary Material). Eligible participants (ECAMPs) as defined as per operation criteria were identified (sample of convenience) across different regions of the world using membership directories of professional societies in the field and social media/research networks, that is, ResearchGate and LinkedIn. ECAMPs ($n = 270$) were then invited to participate in the study via email. Subsequently, the data were analyzed, and the recommendations were compiled based on feedback from a core group of 13 collaborators of the ISAM NExT expert committee for the research project.

### Inclusion and exclusion criteria

For the purpose of this study, the operational definition of "ECAMPs" has been used (Table 1) who were clinicians, scholars, resident doctors, and professionals working in or with an interest in the field of addiction medicine within 10 years of obtaining MD/MSc/equivalent degree *or* within 5 years of obtaining PhD degree depending on national context and were aged between 25 and 45 years.

All participants who gave informed consent to participate in online surveys and expert group discussions were included in the study. A sample size of a minimum of 100 respondents for phase I (online Google Form$^{LM}$ survey) across at least 10 countries in the world and a minimum of 20 respondents from phase I to phase II (online expert group discussion) were decided based on feasibility, time constraints, and their COVID-19 pandemic circumstances (Rayhan et al., 2013).

### Recruitment strategy

For the first phase of the study, potential participants were identified using the membership directory of organizations working in the field of addiction medicine (e.g., ISAM, International Society of Substance Use Prevention and Treatment Professionals [ISSUP], World Psychiatric Association [WPA], Indian Psychiatry Society [IPS], and social media [LinkedIn] and research network [ResearchGate]). We ensured that our sample included at least 10 participants from each World Health Organization (WHO) region (African Region, Region of the Americas, South-East Asia Region, European Region, Eastern Mediterranean Region, and Western Pacific Region) in order to increase diversity and global representation. For the second phase of the study, all participants who took part in the first phase were randomized as per WHO regions and were invited to participate in in-depth interviews (within online expert group discussions) on the themes that emerged on training, research, and mentorship, using stratified random sampling strategy. A random sample of participants was engaged in three sessions of discussion, each comprising 8–11 respondents, for a duration of 2 h, in April 2021. The participants were contacted in advance through email with enclosed information about the questionnaire, an expert group discussion guide, the rules of engagement in the discussion, the participant information sheet, and a consent form. Upon receiving consent, a link for an online meeting was shared.

The online expert group discussions (training, research, and mentorship) were facilitated by collaborator members from the ISAM NExT. The moderator (a trained ISAM NExT Member) guided the participants with questions and facilitated the discussion. The meetings were video recorded (with permission from the

**Table 1.** Phase I survey – Socio-demographic and addiction medicine training-related information ($n = 110$)

| Variables | Mean (SD)/ frequency (%) |
|---|---|
| Age (in years) | 35.66 (4.97) |
| Gender | |
| Male | 68 (61.8%) |
| Female | 42 (38.2%) |
| World Health Organization (WHO) regions | |
| Africa | 27 (24.5%) |
| Americas | 14 (12.8%) |
| Eastern Mediterranean | 12 (10.9%) |
| European | 16 (14.6%) |
| Southeast Asia | 28 (25.4%) |
| Western Pacific | 13 (11.8%) |
| Type of professionals | |
| General/addiction psychiatrist | 62 (56.5%) |
| General/addiction physician | 30 (27.3%) |
| Clinical psychologist | 5 (4.5%) |
| General/psychiatrist nursing | 4 (3.6%) |
| Psychiatric social worker | 4 (3.6%) |
| Doctoral (PhD) student/post-doctoral fellow | 5 (4.5%) |
| Years of clinical/research experience in addiction medicine | 5.78 (9.35) |
| Deficiency in existing addiction training program/ curriculum for ECAMPs | Yes – 95 (86.4%) No – 15 (13.6%) |

participants) and later transcribed for thematic/content analysis (Elo and Kyngäs, 2008).

### Ethics approval and consent

The online survey was conducted according to the principles of good scientific practice (Eysenbach, 2004). Ethical approval for the study was sought and granted by the Institutional Ethics Committee (IEC) at the All India Institute of Medical Sciences (AIIMS), New Delhi, India (September 16, 2020, reference number IEC-888/04.09.2020). Participants provided a written online informed consent to participate in this study before filling out the self-administered survey, voluntary and anonymously.

### Analysis

Data of phase I were analyzed using the Software Package for Social Sciences for Windows v. 24.0 (SPSS 24) (IBM Corp, Armonk, NY, USA). Categorical variables were summarized as $n$ (%), and continuous variables as means (standard deviation [SD]). Focus group discussion data were analyzed using content analysis of the video-recorded expert group discussions transcribed to the word document by the independent researchers (P.K. and K.R.). A core group of 13 collaborators from the ISAM NExT committee reviewed the data before publication.

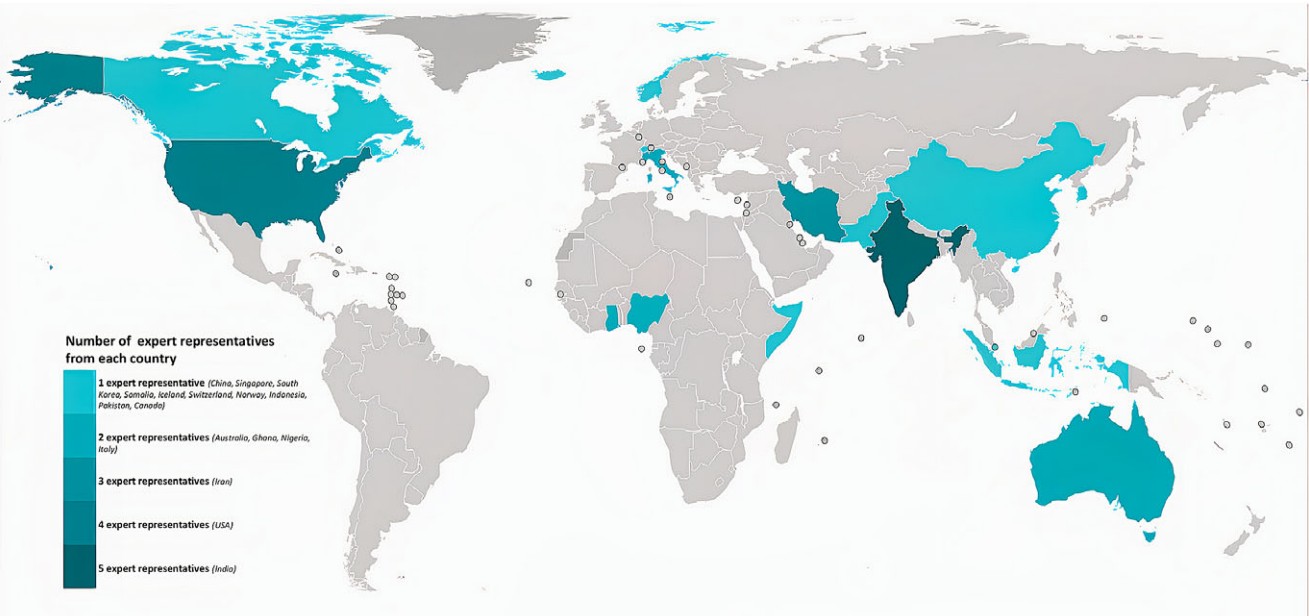

**Figure 1.** The number and geographical distribution of representatives contributed to expert group discussions illustrated on a world map.

## Results

### Phase I: Online survey

Out of the 270 potential respondents approached, a total of 125 responses were received from across 56 countries and 6 WHO regions during 6 months of the data collection period (response rate 46.3%). Fifteen responses were excluded as 10 respondents did not fulfill eligibility criteria and 5 responses were duplicates; therefore, data were analyzed for a total of 110 respondents for phase I of the survey (Table 1). The mean age of respondents was 35.66 (SD = 4.97) years. About half of the respondents were from the Southeast Asia and Africa region. Respondents had been working in the field of addiction medicine or psychiatry profession for an average of 5.78 years (ranging from 1 to 17 years). Out of these respondents, 56% were psychiatrists, with some exclusively practicing addiction psychiatry. Around 27% of respondents were addiction medicine physicians; psychologists (4.5%), nursing professionals (3.6%), social workers (3.6%), and post-doctoral research fellows (4.5%) constituted the remaining professionals. Of the 110 respondents included, 86.4% identified deficiencies in existing addiction training programs and curricula for ECAMPs. Key themes for addiction medicine training needs were identified including a recommendation for trained addiction professionals as mentors (31.8%), networking opportunities (17.3%), institutional support, and having a mentorship program as an integral part of training. Ways to improve upon research opportunities for ECAMPs were identified including compulsory research projects (28.6%), workshops on conducting research (26.5%) and obtaining funding/grants (20.4%), funding (16.3%) and interdisciplinary teams (8.2%). Lack of research opportunities, exposure to specialty clinics, trained faculty, standardization, and less exposure in the clinical realm were all identified as gaps for ECAMPs.

### Phase II: Online expert group discussions (n = 28)

All participants of the phase I study were invited to participate in group discussions on training, mentorship, or research needs. A total of 110 participants were randomized based on the six WHO regions (stratified) into groups of 37, 37, and 36. Subsequently, these groups were assigned one of the three themes (training, mentorship, and research), and an email invitation along with questions on that theme was sent 10–14 days before the online expert group discussions. A total of 28 respondents participated in phase II (overall response rate 25.4%). The response rate was 8/36 (22.2%) for the training theme, 8/37 (21.6%) for mentorship theme, and 12/37 (32.4%) for the research theme. The geographical distribution and the number of representatives in expert group discussions are depicted in Figure 1. Of the 110 respondents included, 86.4% identified deficiencies in existing addiction training programs and curricula for ECAMPs. Key themes for addiction medicine training needs were identified including a recommendation for trained addiction professionals as mentors (31.8%), networking opportunities (17.3%), institutional support, and having a mentorship program as an integral part of training. Ways to improve upon research opportunities for ECAMPs were identified including compulsory research projects (28.6%), workshops on conducting research (26.5%) and obtaining funding/grants (20.4%), funding (16.3%) and interdisciplinary teams (8.2%). Lack of research opportunities, exposure to specialty clinics, trained faculty, standardization, and less exposure in the clinical realm were all identified as gaps for ECAMPs.

The content analysis results of the expert group discussions are as shown in Tables 2–4.

### Expert group discussion on training (n = 8)

Expert group discussion on training was participated by respondents from Italy (2), India (2), Ghana (2), Indonesia (1), and Iran (1). We used a structured questionnaire to understand about status of addiction training in participating countries before initiating expert group discussion. The phases of addiction training of participants were postgraduate/super-specialty trainee resident (3/8), junior faculty member/consultant (3/8), and nursing professional (2/8). Most respondents (6/8) reported there is standard training in addiction medicine/psychiatry in their

**Table 2.** Content analysis of expert group discussion (training)

| Expert group discussion | | |
| --- | --- | --- |
| Question 1 (Q.1). What do you think about issues faced by early-career addiction medicine professionals in terms of standardized training, deficiencies, suggestions for improving the same? | | |
| Themes | Examples | Remarks/consensus |
| Entry/eligibility to join addiction medicine training/ course | *"Both physicians and psychiatrists can practice addiction medicine in Italy."* General psychiatrists manage addiction problems in Iran. The eligibility for training is defined and limited to only psychiatrists in India. Only a few institutes offer addiction medicine training in Indonesia and India. *"The training is heterogeneous and varies across institutes in India."* There is an MCQ-based entrance examination for entry into addiction training in India. Training is available for nursing professionals in addiction treatment. | No national training programs in addiction medicine in Italy and Ghana. *Consensus: There is a variation in entry into the addiction training program across countries.* |
| Q.2. What is the duration of posting/rotation in the addiction medicine/psychiatry in your respective countries? | | |
| Duration of posting/rotation in addiction medicine training | No exposure for undergraduate students. Postgraduate students are posted for 3–6 months. *"The training duration varies from 1 to 3 years and super-specialty courses available in India where trainees are posted for 21–14 months in core specialty and remaining months in Neurology, Gastroenterology, Child Psychiatry, etc."* There is no specialized training in Italy, and psychiatry residents are posted for 3 months to 2 years. There is a 3-month duration of posting for a psychiatrist in addiction medicine in Iran, which is not enough. *"There is no rotation in addiction medicine in Ghana, and there are no trained addiction medicine doctors."* In Indonesia, there is a 3–4 weeks posting for undergraduate medical students and 1 month posting in addiction psychiatry for psychiatry residents. *"Two years of experience in the addiction treatment facility is needed to practice as a specialist."* Recently a subspecialty program in addiction medicine started in one university in Indonesia. | Addiction medicine training was recently established in Indonesia, and the curriculum is yet to develop. *Consensus: There is a variation in the duration of training/posting in addiction medicine across countries.* |
| Q.3. What about nature of addiction medicine/psychiatry training in your respective countries? | | |
| Nature of training | An adequate number of patients with substance use disorders across countries for training needs. Both buprenorphine and methadone are available as opioid agonist treatment (OAT) in Italy, India, and Indonesia. In Iran, tincture opium is also available for treatment. *"There is also a separate 3-month course in opioid agonist treatment including methadone for physicians in Iran."* Treatment options are limited to tramadol in Ghana. Medications are available for the treatment of alcohol use disorders in all countries. In-patient treatment facilities are available in all countries. | ECAMPs are exposed to clinical training in outpatient treatment services in all participating countries. *Consensus: Outpatient treatment facilities are key places for clinical exposure.* |
| Q.4. What about exposure to various specialties of addiction medicine/psychiatry? | | |
| Exposure to specialty clinics/ sub-specialties | *"In Iran, exposure to various substance use problems is diverse, and supervision by expert faculty is available."* Specialty clinics are still in the budding stage across countries, and some countries have behavioral addiction clinics. In Italy, trainees are able to see dual diagnosis patients as a part of a training program. No NPS clinic in India; other specialty clinics are available. No specialty clinics in Ghana. In Indonesia, exposure to behavioral addiction and tobacco cessation clinics are available as a part of training in limited institutes. | Trained faculty in addiction medicine are available in Iran. *Consensus: There is limited exposure to specialty clinics across countries.* |
| Q.5. What about assessments during addiction medicine/psychiatry training in your respective countries? | | |
| Assessment at the end of the training | Six monthly assessments of training in Italy. End semester comprehensive exams, including theory and clinical case presentation, is conducted in India for super-specialty trainees. No separate assessment in Iran and Indonesia for MD (Psychiatry) residents. *"10% marks of Objective Structured Clinical Examination (OSCE) are dedicated to addiction psychiatry in the MD (Psychiatry) final exam in India."* In India, the assessment is predominantly summative with some elements of formative assessment. Clinical assessments are conducted for nursing professionals specialized in addiction treatment in Ghana. | End of the training standard assessment is conducted in India. *Consensus: Training assessments are conducted in all the countries. However, the method of assessment varies.* |

respective countries. There is no addiction training program in Ghana. The average duration of training is 3 years on less. The majority of respondents were satisfied (4/6) with the available training program in their countries.

The main strengths of the training program are exposure to the management of different substances/specialty clinics (4/6) and the availability of eminent addiction professionals as mentors (2/6). While main limitations of the training program are the lack of addiction specialists as trainers (4/6), lack of specialty clinics and exposure to multiple substance use problems (1/6), and difficulties in obtaining needed materials for training (1/6). The majority of

respondents (4/6) reported that on an average trainee consults more than 50 patients/year.

All respondents reported that there is exposure/posting/rotation to addiction medicine/psychiatry during medicine or psychiatry training. Duration of posting varies from 1 to 6 months. There is exposure to special treatment clinics, for example, dual diagnosis clinics, opioid agonist treatment clinic, behavioral addiction clinic, and consultation liaison clinic in the majority of countries (7/8). Community addiction treatment clinic, adolescent clinic, and tobacco cessation clinic posting are available in (4/8) countries. Only one country, that is, Italy has a clinical posting in the pain

**Table 3.** Content analysis of expert group discussion (mentorship) (*n* = 8)

| | Expert group discussion | |
|---|---|---|
| **Question 1 (Q.1). In your university, in your city, in your academic field, do you have a mentorship program in the field of addiction medicine or general psychiatry? If so, how satisfying is the program?** | | |
| Themes | Examples | Remarks/consensus |
| Availability of mentorship program | *"There is no formal mentorship program in some institutes. But anyone can identify someone with similar research interests as a mentor. There is a mentorship program for buprenorphine training to get a special waiver to prescribe buprenorphine in USA (called as Prescribers' Clinical Support Systems for Opioid Therapies [PCSS-O]), but it is not associated with any university, and it is not that satisfying."* | Mentorship programs for early-career addiction medicine professionals are not available in most countries. *Consensus: The mentorship program is not available widely in developed countries and non-existent in developing countries.* |
| Nature of mentorship program | *"In Canada, there is mentorship program only for current fellows in addiction medicine, for the duration they are in the program to provide them primarily with career advice."* | *Consensus: There are no formal mentorship programs for addiction medicine in most institutes, and among the available programs, only a few are satisfactory.* |
| **Q.2. Can you describe reasons for effectiveness and non- effectiveness for mentorship programs?** | | |
| Effectiveness/non-effectiveness of the mentorship program | *"Mismatch about expectations is a very critical thing. There should be flexibility in the program for people or prospective mentees to be specific in what they are looking for and choose mentors based on their expectations. The relationship between mentor and mentee is another important reason for failure or success of a program."* The program should be interdisciplinary, and mentees should be able to choose their mentor based on their needs or interests; The mentor should have adequate emotional intelligence, should be available and accessible, and be able to relate to the mentees, and there should be specific, measurable, achievable, relevant, and time-bound SMART goals. The program should not be short-term but should be continuous or long-term. | *Consensus: For a successful mentorship program, it should be introduced at early-career level, should be flexible keeping mentees in the left of everything, there should be SMART goals.* |
| **Q.3. What are your suggestions for successful mentorship programs, especially for early-career addiction medicine professionals?** | | |
| Appropriate time and duration for the mentorship program | *"Exposing medical students early in their career, preferably at the undergraduate level even before they decide to go into psychiatry or surgery specialties,* etc. *Identifying those who are passionate about addiction. For graduates or those in private practice,* etc., *International Society of Addiction Medicine (ISAM) or medicine supported recovery (MSR) can create networks. There is a need for continuity instead of being short-lived. Information about mentorship programs should be given to trainees during the last year of training."* | |
| Career-based mentorship program | *"We need to have early-career mentoring programs. Successful mentoring programs should have time-bound SMART goals. Also, some kind of career-based mentoring on how to maintain a balance between personal and professional aspects of life. We should have some wellness programs on how to balance lifestyle and work in addiction."* For a successful mentorship program, it should be introduced at the undergraduate level, postgraduate residency as well as early-career level. There should be flexibility keeping mentees in the left of everything, and mentees should be able to choose their mentors. There should be SMART goals. The program should not be short-lived but should be continuous and comprehensive. ISAM can help in international collaboration for mentorship. | Continuous mentorship program as per the stage of career is needed for its effectiveness in shaping the career of ECAMPs. *Consensus: There should be a career-based mentoring program to discuss work–life balance issues.* |

clinic. The content analysis of expert group discussion is shown in Table 2.

### Expert group discussion on mentorship (n = 8)

*The expert group discussion on mentorship was participated by eight respondents from seven countries, that is,* Canada (1), Iran (1), India (1), Nigeria (1) Singapore (1), Somalia (1), and USA (2). We used a structured questionnaire to understand about status of mentorship program in participating countries before initiating expert group discussion. The proportion of participants in the different phases of careers were postgraduate/super-specialty trainee residents (3/8), junior faculty members (2/8), doctoral/fellowship students (1/8), post-doctoral student/fellow (1/8), and associate physicians (1/8). Only 3/8 participants were informed that there is a formal mentorship program in their respective countries.

About (5/8) of respondents reported that they consider their current supervisor/thesis guide/principal investigator as a mentor. A participant informed during a survey that she has a long-distance mentor from outside her university. Only one respondent (1/3) reported that he is satisfied with the existing mentorship program. The reason for the effectiveness of the successful mentorship program was similar experiences, life goals, and professional aims to the respondent, an ability to relate (2/3), and a genuine concern for the respondent's wellbeing and success (1/3). While the reason for the non-effectiveness of the mentorship program was limited time and busy schedule, lack of protected time for mentoring and research (2/3), and mentors possessing biased views against non-

**Table 4.** Content analysis of expert group discussion on research needs (*n* = 12)

| Question 1 (Q.1). What is the scope of research in addiction medicine in your country? | | |
|---|---|---|
| **Themes** | **Examples** | **Remarks/consensus** |
| Research themes covered worldwide | *"In Korea, many are interested in research in gaming disorders and other behavioral addictions. The government encourages the researchers to do various pieces of research in the field of behavioral addictions and alcohol use disorder."* China and Korea are more focused on behavioral addictions. In Australia, research focus is on psychedelics, methamphetamine, amphetamine-type stimulants, newer psychoactive substances, as well as tobacco and alcohol. In USA and Iceland, all types of illicit substances are focus for research. In Nigeria, there is a need for research in addiction. | *Consensus: Heterogeneity in the scope of research in addiction medicine among countries.* Research topics orientation based on a specific drug or behavior. |
| Determinants of research themes conducted | Cultural influences (politics, religion) on research areas investigated. USA: "Research on all types of drugs are being conducted and all varieties of work including controversial researches with political and religious influences." Economic influence (funding opportunities) on the scope of research themes investigated. Iceland: "There is a small population, but accessible and adequate funding opportunities for research." | |
| Q.2. Are there enough research opportunities, either funded or non-funded, for early-career addiction medicine professionals in your institution? | | |
| Impact of funding on research opportunities for early careers in addiction medicine | Inadequate funding for addiction research compared to other psychiatric disorders and non-communicable diseases. In Australia, there are enough research opportunities. In China, there is difficulty in getting funding for research, although the government is advocating for conducting research on behavioral addictions as it is a public health problem. In the USA, Iceland, and Norway, plenty of funding opportunities are available. "There is no sufficient funding for research in Nigeria, Pakistan." | Some jurisdictions, totally (e.g., African countries) or partially (e.g., China) resulted in a lack of opportunities for research in addiction medicine. *Consensus: The funding opportunities varies across countries.* |
| Impact of mentorship on research opportunities for early careers in addiction medicine | Nigeria: *"There are no sufficient mentorship opportunities for research."* Some mentorship opportunities are available in most other countries. | |
| Impact of clinical workload on research opportunities for early careers in addiction medicine | USA: *"It is difficult to manage time between research and clinical work."* | |
| Q.3. How independently the research (meaning not funded by industries with competing interests like alcohol, tobacco, gambling, and gaming industry) could be conducted in your country? | | |
| Availability of infrastructure for independent research | *"I have never heard about competing for interest institution ever participating or having influence in addiction research in Iceland. We have access to independent research."* | *Consensus: Conducting research projects independently is possible in most countries.* |
| Q.4. How easy is it for you to consolidate research, clinical, and training activities? | | |
| Multitasking | In Australia, it depends on papers and position (academic/clinical/combined). It is easier to combine research activities with training activities. However, it is harder to make systematic reviews or other things like that during business hours. It is easier to do teamwork. | *Consensus: It is difficult to combine clinical work with research work in many countries given the time, energy burnout, etc.* |
| Q.5. How do you manage time, in between clinical and training work for research work, in terms of applying for funding, writing protocol, research projects, etc.? | | |
| Dedicated time for research | *"It is difficult to manage clinical and training work for research work. I do not have the expertise or skills for applying for a larger-scale grant – mentorship and training are needed for this purpose. Also, I have to spend more time outside my clinical and teaching hours. It is a struggle in terms of balance." A* | *Consensus: It is easier to be part of a research group with share responsibilities rather than doing independent research.* |

(*Continued*)

**Table 4.** (*Continued*)

| Themes | Examples | Remarks/consensus |
|---|---|---|
| | *respondent from the USA.*<br>In the USA, it is a struggle to manage timings for clinical and training work for research. In Australia and China, grant writing is the biggest time suck – We put in a lot of effort outside usual working hours, but in the end, you may not get funding. | |
| Q.6. Do you have a form of a research program in your medical school/institute/university for a research program in addiction medicine? And if there is, are you satisfied with the program? (very unsatisfied to very satisfied) | | |
| Availability of research program | In Australia, there is a research program as part of specialty training in addiction and PhD opportunities in addiction. Research activity satisfaction depends on your personal interest in research or the project. In the USA, the satisfaction of research programs depends on the location of the practice. Some people say that it is pretty satisfying. In China and Korea, there are satisfying research programs.<br>In Nigeria and Iceland, there is no addiction psychiatry research program. | *Consensus: Research program for ECAMPs are available in developed countries, e.g., USA/Australia.* |
| Q.7. Can you describe any two reasons for the effectiveness or non-effectiveness of the available research programs you just described in your country? | | |
| Effectiveness of mentorship program | *"The primary reason for effectiveness as well as non-effectiveness of Australian research program is the fact that it is mandatory. It is effective because you have to do it. It is ineffective as it is forcing people into research, when they may be not interested at that time of training, producing opposite effects."*<br>In Australia, research is mandatory in research programs. This is a reason for effectiveness as people do it, as well as non-effectiveness, forcing people who may not be interested in doing research at that point in life.<br>In the USA, the reason for the effectiveness of research usually involves specific people and when specific research is done. The reason for non-effectiveness is that it is difficult for people to enter into research from clinical work.<br>In China and Korea, the programs are free-styled and depend on the style of mentors, which is the reason for non-effectiveness. There is a need for standardization or regularization for the program and the mentors.<br>In Iceland and Nigeria, there is no program, and there is a need for the same. | *Consensus: Effectiveness of research program is subjective to mentors. Research as built in part of clinical, training project is more effective.* |
| Q.8. Would you be interested in workshops on scientific work? And if any, what would you be interested in (workshop on finding donors, scientific writing, or any other topics)? | | |
| Modes of improving research program | In Australia, there is a need for workshops on the structure of research papers, writing papers in an efficient way, and grant writing. Some didactic teaching could be repetitive as universities often have online guides or webinars on academic writing, literature review, etc.<br>In the USA, workshops are not necessary, as most programs have research-related guidance built-in. There is a need for pairing with a mentor and support groups for paper rejection.<br>In China, the participant wanted training on the whole process of publication – how to get a paper published, how to cope with every step, and how to cope with rejection.<br>In Korea, there was a need for workshops on the structure of research paper.<br>In Nigeria, there was a need for finding suitable journals. | There is demand for workshops on scientific writing, obtaining grants among ECAMPs.<br>*Consensus: Mentorship for research paper writing for ECAMPs is needed in most countries.* |

**Table 4.** (*Continued*)

| Themes | Examples | Remarks/consensus |
|---|---|---|
| Q.9. Is there an addiction research group community or network where you work which would be interested in participating in digital meetings with other PhD students, postgraduates, etc., in other countries? | | |
| Professionals groups engaged in research | In Australia, the USA, China, and Korea, addiction research groups or communities exist. All expressed their interest in participating in digital meetings. In Iceland, there is an addiction research group and community. The representative stated that she was interested in participating only when she got into a residency program or similar qualification, as she felt it would be pointless without active interaction. | |
| Q.10. What are your suggestions for the improvement of research activities in the field of addiction medicine? | | |
| Ways to improve research opportunities | *"It would be wonderful if there was a mechanism that can provide some guidance or huge pass for research. One association of international organization can provide the junior researchers with a road map for research – what we can do, what we can research on, what we can start – More than mentorship – We can gather a lot of eminence or important researchers, cutting edge directions for research in the future for next 5 years or 10 years"* (Respondent from China) In Australia and the USA, having a website or page or resource listing opportunities can help to collaborate and/or work regionally as well as internationally. In China and Korea, a mechanism to provide guidance or huge pass for research and organization of training provides more resources and more platforms for junior researchers who do not speak English. Mentorship program is needed. In Iceland, there is need for international collaboration. In Nigeria, there is a need for workshops for writing and applying for grants and international collaboration. | There is a need for more funding for ECAMPs and a roadmap for developing/improving their research skills. *Consensus: International collaboration is key for improving research activities in the field of addiction medicine/psychiatry.* |

psychiatrist's ability to practice in the field of addiction (1/3). Most respondents reported that a single mentor is a good idea (5/8), however, a respondent also expressed that multiple mentors with a single senior mentor in charge (1/8) is a better idea. Respondents informed that as per (Zachary's The Mentor's Guide, 2000) most important phases of the mentorship program in order were: (1) cultivation or negotiation and enabling or protégé (5/8), (2) initiation or preparation (1/8), (3) separation or closing or break up (1/8), and (4) redefinition or lasting friendship (1/8).

Only one respondent reported having an office for delivering needed guidance on career development/training/post-doctoral while training in the medical school/institute/university. Respondents expressed a need for guidance and assistance from mentors in order (1) long-term career planning (3/8), (2) research design (1/8), networking nationally and internationally (1/8), (3) balancing personal and professional demands (1/8), (4) developing a research portfolio (1/8), and (5) addressing burnout during training (1/8).

Respondents expressed that by the end of an effective mentorship program, the following abilities a mentee should obtain in order (1) ability to conduct research ethically independently and responsibly in the given area (4/8), (2) ability to achieve career progression and financial independence through satisfactory job opportunities in the given area (3/8), and (3) ability to become a mentor and run an effective mentorship program in the given area (1/8). The content analysis of the expert group discussion is given in Table 3.

*Expert group discussion on research needs (n = 12)*

Expert group discussion on research needs was participated by respondents from 11 countries including Australia (2), China (1), Iceland (1), India (2)*, Iran (1), Nigeria (1), Norway (1), Pakistan (1)*, South Korea (1), Switzerland (1), and USA (2). Two participants left the group discussion halfway, 12 participants attended the full discussions. The content analysis for expert group is shown in Table 4

## Discussion

This study was one of the most extensive surveys conducted among ECAMPs assessing the need and scope for standardized training, mentorship programs, and research opportunities — the online survey methodology allowed for a broad representation of participants from 56 countries.

There is wide variability in entry requirements for addiction medicine training globally. In countries such as the USA and the UK, both family physicians and psychiatrists can practice as addiction specialists. In others, such as in Italy, both physicians with a specialty in internal medicine or other medical sub-specialties together with psychiatrists and pharmacologists can treat patients with addiction. In most other countries, only psychiatrists can train in addiction medicine as a specialty. Developing countries allow nursing practitioners and social workers to pursue addiction medicine training due to a shortage of specialty physicians.

Training and exposure to addiction medicine also differ considerably across different countries. With respect to undergraduate training, exposure to addiction medicine as a specialty is minimal (about 1 week in the USA, 7 h in the UK) or absent in most countries (Iannucci et al., 2009; O'Brien and Cullen, 2011; Tontchev et al., 2011; Ayu et al., 2017; Tripathi et al., 2020). The biopsychosocial model of addiction is taught as a part of the theory in undergraduate medical school (Carroll et al., 2014). Addiction medicine is an integral part of psychiatry and family medicine residencies in some countries like the USA, India, and Iran with respect to postgraduate training. Indonesia and India offer additional certificate courses after postgraduate training. There is considerable variation in curriculum and duration of the training placement of psychiatry residents in addiction psychiatry across countries.

In the USA and Australia, addiction psychiatry is a separate specialty as a postgraduate program. In India and Indonesia, the addiction psychiatry specialty is in its early developing stage (Pinxten et al., 2011; Das and Roberts, 2016; Tripathi et al., 2020). The lack of a structured curriculum is an important issue highlighted by the present survey participants. The content of the addiction medicine curriculum varies due to the nature of local substance use, availability of specialty clinics, opioid agonist treatments and other pharmacotherapy options available in different countries, and availability of trained faculty members for teaching. A structured curriculum can improve the knowledge of addiction medicine among internal medicine residents and hence need to be developed and updated in different countries in order to improve the delivery of quality addiction treatment services (Brown et al., 2013). The quality of training is an issue for both internal medicine residents in the USA and also among psychiatry trainees across European countries as per a recent survey, which makes the call for a structured curriculum ever more important and urgent (Wakeman et al., 2013; Orsolini et al., 2021).

A survey in China revealed that doctors involved in drug treatment are not well prepared or experienced and have negative attitudes toward substance use disorders and afflicted patients (Tang et al., 2005). The low number and level of professional addiction experts are the potential outcomes of inadequate addiction medicine training for medical students and residents in the USA, which is highlighted previously (Rasyidi et al., 2012). From a trainee point of view, there is a demand for standardized training as emphasized in past reviews and found in the present study (Klimas et al., 2020). The evaluation of standardized, structured short-term training is also found to be an effective tool for addiction medicine training (Barron et al., 2012; Ayu et al., 2015). The next generation of addiction treatment providers needs to be trained adequately to deal with emerging substance use problems across the globe.

There is a standard exit exam after completion of addiction medicine training in some countries like the USA and India. There is no exit exam in Australia. There are regular mid-term assessments (6 monthly or yearly) that are also conducted in countries like Italy, India, Iran, and Ghana for the trainees. In this regard, efforts by the ISAM to successfully conduct International Certification in Addiction Medicine exams for global trainees for the past 10 years need to be acknowledged (el-Guebaly and Violato, 2011).

The availability of mentorship programs and needs were assessed in the present study. We found there are limited mentorship programs available for ECAMPs. Such programs are limited to developed countries like the USA and Australia. The mentorship program is non-existent in most African and Asian countries like Ghana, Nigeria, China, India, Indonesia, etc. Most participants recognized their training program supervisor and thesis advisors as a mentor. A single mentor was desired by most, although some participants expressed the need for multiple mentors depending upon the need in particular areas of interest and the stage of their career. The barriers identified for quality mentorship programs were lack of time, funding, and trained faculty members (Kahan et al., 2001). Most participants favored a continuous mentorship program in different stages of their careers and were not limited to only the training duration. Mentorship programs are vital for the development of the career of ECAMPs, and there is a need to facilitate mentorship programs across countries as reported by participants the importance of mentorship program is also highlighted by NIDA (National Institute on Drug Abuse (NIDA), 2018). There are limited research studies on understanding the challenges faced by mentorship problems for ECAMPs. Among the available programs, The Learning for Early Careers in Addiction & Diversity (LEAD) Program, funded by the National Institute of Drug Abuse, uses a team mentoring approach. Each LEAD Program scholar works with a Clinical Trial Network (CTN) primary mentor while also receiving guidance from a University of California San Francisco (UCSF) mentor and a nationally regarded diversity advisor (UCSF, 2020). Other similar programs are run by addiction medicine societies like the American Academy of Addiction Psychiatry (AAAP) and ISAM (Academy of Addiction Psychiatry (AAAP), n.d.). There is a tremendous need to develop a mentorship culture to strengthen academic medical centers engaged in addiction medicine training. Innovative methods like co-training with general physicians can facilitate mentorship programs in such centers. The mentoring need is now even greater with the expansion of addiction medicine as a specialty and many young professionals joining their respective training programs (Alford et al., 2018; Choi et al., 2019; Academy of Addiction Psychiatry (AAAP), n.d.).

Most of the study participants reported there are limited research opportunities for ECAMPs. There are many challenges like clinical workload, funding, few suitable research mentors, obtaining research grants, and publishing the research. The challenges are existent even in developed countries like the USA and Australia. The research capacity has to be more developed during the training program and is effective when mandatory for the completion of training. There is an unmet demand for grant writing, workshops for conducting research, and writing papers among ECAMPs. The research grants available from NIDA are mostly limited to USA residents/citizens (National Institute on Drug Abuse (NIDA), 2020). There are limited opportunities in addiction medicine societies. The United Nations Office on Drugs and Crime (UNODC), with support from the Drug Abuse Prevention Center (DAPC), started offering grants for early-career researchers for projects related to prevention and promotion activities recently (United Nations Office on Drugs and Crime, n.d.).

Combining clinical training and research would be a step ahead in improving addiction medicine training programs and creating research opportunities for ECAMPs (Klimas et al., 2017). Developing research capacity among ECAMPs from low-income countries and LMICs by conducting workshops with the support of facilitators from HICs can be a solution for the problem. Other allied addiction medicine professionals can also be engaged in such training programs to develop the workforce and build more capacity (Merritt et al., 2019; McCarty et al., 2020; Masson and Sorensen, n.d.). The

main challenges encountered in conducting research by ECAMPs in the European survey (*n* = 258) were lack of time as a large proportion of participants (87.2%) reported conducting research after regular working hours or partly during and after working hours. Only one-tenth ever received a grant for their work. Lack of funding is an important hurdle in conducting research in spite of ECAMPs being motivated to conduct the research (Koelkebeck et al., 2021). Global societies and institutes working in the field of addiction medicine need to provide adequate research opportunities as there is a risk of ECAMPs falling prey to predatory publishing and industry-sponsored research in the early stage of their career, which may bias their subsequent research projects (Bhad and Hazari, 2015; Forero et al., 2018; Mitchell and McCambridge, 2021).

The results from the present study suggest that there is variation in eligibility, the content of the curriculum, and assessments for addiction training across the globe. It is essential to develop a standard curriculum and training content that is competency-based, culturally sensitive, and can include local jurisdictional norms with substance use disorders. Flexibility is needed in the curriculum to account for the possibility of various medical professionals starting addiction medicine as a career. The study findings emphasized the need for mentorship programs and more research opportunities for ECAMPs as a vital component of addiction medicine training.

A major strength of the present study is the perspective from more than 50 countries and covering all six WHO regions. We used a robust methodology with an online two-phase survey with systematic randomization for the second phase. The second phase, that is, the qualitative part of the study using expert group discussions, adds perspective on the attitudes and opinions of survey participants and hence adds more meaning and depth to the data collected using the online survey. Limitations of our study include self-reported data and relatively small sample size. The study was conducted during the COVID-19 pandemic when there was a disruption in training programs and a shift to online teaching, which may have influenced some of the findings in the study. The generalizability of the data is another limitation, as only participants who were members of professional societies and were available on professional social media platforms were approached. Future studies should address these limitations using randomized control trials for studying models of training, innovative techniques of training, and longitudinal study design to study mentorship needs in long-term career growth.

The study findings emphasize the need for standardized training programs, improving research opportunities and collaboration, and effective mentorship programs for the next generation of addiction medicine professionals. We propose following recommendations based on findings of the present survey.

a. The lack of standardized training in addiction medicine across countries is a major issue, so the training gap should be assessed using standard measure across countries.
b. As the goal of standardized training in addiction may not be achievable in all countries, the training programs could be supported, supplemented by global societies, organizations including World Health Organization (WHO), NIDA, and ISAM to address the training need.
c. There is need for mentorship program among ECAMPS across countries.
d. The effective global mentorship program in addiction medicine which is culturally competent, accessible needs to be developed by global societies, organizations including World Health Organization (WHO), NIDA, and ISAM working in the field.
e. There is need to increase research opportunities for ECAMPs across countries by expanding research scholarships, grants targeting the group.

## Conclusions

The present global survey by ISAM NExT is one of the few studies that assessed the training needs, research and mentorship opportunities among ECAMPs. The report highlights deficiencies in standardized training and assessment, lack of research and mentorship opportunities for the group. It is important to address the gaps in training and nurture next generation of addiction medicine professionals by providing adequate research and mentorship opportunities. Global workforce development is a key for mitigating emerging post pandemic challenges in the field of addiction medicine.

## Members of ISAM NExT Consortium

Members include 1. Ankita Chattopadhyay; 2. Bezzina Gianluca; 3.Chia-Chun Hung; 4. Danielle Jackson; 5. Edem Sallah; 6. Enjeline Hanafi; 7. Erna Gunnthorsdottir; 8. Francina Fonseca 9. Gayatri Bhatia; 10. Georgios Tzeferakos; 11. Hillary Selassi Nutakor; 12. Hussein Elkholy; 13. Irfan Ullah; 14. Jiang Long; 15. Jibril Handuleh; 16. Leonardo E.Allagoa; 17. Mandana Sadeghi; 18. Mehrnoush Vahidi; 19. Mohammed Aljenibi; 20. Mohammadreza Shalbafan; 21. Paxton Bach; 22. Sung Young Huh; 23. Surajuddin Abddulkadir; 24. Takeo Toyoshima; 25. Vicky Phan; 26. Wafaa Elsawy.

**Open peer review.** To view the open peer review materials for this article, please visit http://doi.org/10.1017/gmh.2023.35.

**Acknowledgments.** We would like to thank Dr. Shalini Arunogiri, Training Officer, ISAM and Dr. Cor De Jong Co-chair, Training Committee, ISAM; Dr. Susana Galea Singer, Education Officer, ISAM; Dr. Nady el-Guebaly, Chief Examiner, ISAM for their guidance and Ms. Marilyn Dorozio, ISAM office for technical support in conducting online meetings.

**Author contribution.** R.B. and P.K. drafted the protocol. R.B., M.K., P.R., and N.V.L. recruited the phase I survey participants. P.R. sent the invitations and arranged online expert group discussion meetings. S.A., M.F., K.M., and P.R. developed questions and animated the discussions for expert group discussion research theme. R.B., S.A., M.F., K.M., P.R., H.A., J.B., S.T., M.K., L.O., and V.L.N were collaborators for expert group discussions and facilitated the sessions. P.K. was note taker for expert group discussion sessions, transcribed and analyzed the research and mentorship expert group discussions. K.R. validated the expert group discussions data. P.R. illustrated the figure and map. ISAM NExT Consortium members participated in both phases of the survey. M.P., H.E., and A.B. reviewed the manuscript and mentored the group. All authors contributed to the revision and final edits of the manuscript.

**Financial support.** M.F. is supported by NIDA and NIAAA intramural research funding (ZIA-DA000635 and ZIA-AA000218). The content of this article is solely the responsibility of the authors and does not necessarily represent the official views of the National Institutes of Health.

**Competing interest.** The authors have no competing interests to declare.

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
