## [Reviewer Report]

To, 

The Editor, 

Cambridge Prisms: Global Mental Health

Sub: Manuscript titled “An online global survey and follow-up expert groups on the scope and needs related to training, research, and mentorship among early-career addiction medicine professionals” as an original research paper 

Dear Sir, 

I am hereby submitting a manuscript titled “An online global survey and follow-up expert groups on the scope and needs related to training, research, and mentorship among early-career addiction medicine professionals” for exclusive publication in your esteemed journal. The manuscript has been read and approved by all the authors, and the requirements for authorship as stated in the journal’s instructions to authors have been met. I hope you will find the documents in order. 

Roshan Bhad MD

---

## [Reviewer Report]

This study provides an international update regarding training in addiction medicine around the world. This field is indeed still emerging and training varies considerably between countries. The strength of the paper is an effort to sample many countries around the world but not enough of this knowledge has been presented in the results.

Table 3A and 3B – these summarise responses from 8 participants and should be summarised in a few lines of text rather than this lengthy table. Some comments (eg absence of dedicated training programs in a number of countries) are not new knowledge.

Table 5 similarly is extremely long for the degree of content. There is a repeated tendency to describe trends as representing the country of origin when there was literally one person interviewed from each nation.

Discussion states “Mentorship programs are vital for the development of the career of ECAMPs,” this is neither a conclusion of the study, nor is the source or validity of this statement demonstrated. At a minimum, a reference should be provided.

“There is a standard exit exam after completion of addiction medicine training in some countries like the USA, Australia, and India”. This statement is incorrect at least in relation to Australia. i am very familiar with the training scheme and there is no exit exam in that country. The authors should check for all countries and revise accordingly.

I would have liked to see a concise set of recommendations arising from the project, ideally focussed and practical. These could be presented as a short table or as text.

There is relatively little detail for survey results which is a shame as this covered many countries with a significant total sample size. The bulk of the result refer to small ‘focus’ groups where most nations were represented by a single person.

---

## [Reviewer Report]

Overall, I think this is an interesting piece of research that will add to the extremely sparse literature on this topic. The methods were of limited quality and the sample very heterogenous which may limit the impact of the work. Revisions to the methods and results sections would improve the paper.

Title: Suggest using the word focus group to better reflect your methods.

Introduction:

Sentence 1: please provide citation.

The word trainers is used on multiple occasions. This is not commonly used where I am from. Perhaps a better word might be simply mentor, preceptor, faculty members?

Methods: Please clarify why you included early career professionals as opposed to physicians only given the focus on addiction medicine training/competency? I would suggest that you define this early on and provide an explanation of the sample.

Methods: The methods for the focus group are unclear. You write that participants were randomized per WHO regions and were invited to participate. Do you mean that you randomly selected people with representation from each WHO region? Please clarify how you randomly selected people for the focus groups. What if participants declined, did you offer more invitations?

2.4 Please confirm that the written informed consent was inclusive of the focus groups?

Typo, ‘voluntarily’ rather than ‘voluntary’

3.2 I am confused with respect to the WHO regions (3 groups), but earlier you state that they represented 6 WHO regions. Also it sounds like the invitation for the focus group actually went out to all survey respondents, but previously you said they were randomly selected. Please clarify.

Table 3A #10. These categories are not logical. Why was dual diagnosis clinic grouped with opioid agonist treatment and behavioural addiction but separate from community addiction treatment, or tobacco cessation clinic.

Table 3B. Highlighting the consensus statement might be more appropriate than highlighting the ‘remarks’.

For Table 3A and 3B there are a number of acronyms that have not been previously defined.

For the Research discussion was there not a survey circulated in advance like for the other groups?

Table 5 – no ‘consensus’ statement as in the other two tables. The ‘remarks’ section seems to be used for a summary statement

I would report a summary of the qualitative analysis in words in addition to what you have provided in that tables. I find the answers to the survey questions from focus group participants to be distracting to the qualitative analysis. I would clearly present the qualitative analysis results and put the other demographic information in an appendix.

Please format and report the results consistently across the three focus groups.

Overall, I find it confusing to report the interview guide questions alongside the themes. Typically themes may cut across questions as opposed to being direct responses to questions. I do not think this can really be called a thematic analysis. You might characterize this as a qualitative descriptive study using content analysis rather than thematic analysis in my opinion. A citation for the qualitative methods employed might be helpful.

I applaud the authors for taking on a very large project with extremely broad scope.

---

## [Reviewer Report]

Dear Editor,

We have revised our manuscript (GMH-23-0019 ) titled “An online global survey and follow-up expert groups on the scope, training needs, research, and mentorship among early-career addiction medicine professionals” as per the comments by reviewers. We have attached revised version for exclusive consideration of publication in the Cambridge Prisms: Global Mental Health. We thank you for the opportunity. 

Dr Roshan Bhad MD

Corresponding author

---

## [Reviewer Report]

the revisions clarify and improve the manuscript. Its strengths and limitations are evident to the reader.